



**Salinity-depending carbon and nitrogen uptake of two intertidal**
**foraminifera (*Ammonia tepida* and *Haynesina germanica*)**
**Michael Lintner[1], Bianca Biedrawa[1], Julia Wukovits[1], Wolfgang Wanek[2] and Petra**
**Heinz[1]**
**[1]University of Vienna, Department of Palaeontology, Vienna, Austria**
**[2]University of Vienna, Department of Microbiology and Ecosystem Science, Terrestrial**
**Ecosystem Research, Vienna, Austria**
**Abstract**
Benthic foraminifera are abundant marine protists which play an important role in the transfer
of energy in the form of organic matter and nutrients to higher trophic levels. Due to their
aquatic lifestyle, factors such as water temperature, salinity and pH are key drivers controlling
biomass turnover through foraminifera. In this study the influence of salinity on the feeding
activity of foraminifera was tested. Two species, *Ammonia tepida* and *Haynesina germanica*,
were collected from a mudflat in northern Germany (Friedrichskoog) and cultured in the
laboratory at 20 °C and a light / dark cycle of 16: 8 h. A lyophilized algal powder from
*Dunaliella tertiolecta*, which was isotopically enriched with $^{13}C$ and $^{15}N$, was used as a food
source. The feeding experiments were carried out at salinity levels of 11, 24 and 37 practical
salinity units (PSU) and were terminated after 1, 5 and 14 days. The quantification of isotope
incorporation was carried out by isotope ratio mass spectrometry. *Ammonia tepida* exhibited a
10-fold higher food uptake compared to *H. germanica*. Furthermore, in *A. tepida* the food
uptake increased with increasing salinity but not in *H. germanica*. Over time (from 1-5 d to 14
d) food C retention increased relative to food N in *A. tepida* while the opposite was observed
for *H. germanica*. This shows, that if the salinity in the German Wadden Sea increases, *A.*
*tepida* is predicted to exhibit a higher C and N uptake and turnover than *H. germanica*, with
accompanying changes in C and N cycling through the foraminiferal community. The results
of this study show how complex and differently food C and N processing of foraminiferal
species respond to time and to environmental conditions such as salinity.

keywords: benthic foraminifera, feeding experiments, salinity, isotope tracing

**1. Introduction**
The intertidal zone is one of the most extreme habitats on earth. This ecotone, also known as
the foreshore or seashore, is determined by tidal activity. It is an important habitat for various
living organisms like starfishes, sea urchins, corals and foraminifera (Allen 2000). Due to the
alternating presence/absence of water, organisms living here must adapt to the specific
environmental conditions. Important factors shaping the intertidal environment are the
fluctuating water temperature and salinity, pH, available food sources, sediment organic matter
content and fresh water supply. These environmental factors significantly influence the activity



of foraminifera (e.g. Schafer et al. 1996, Caldeira and Wickett 2005, Keul et al. 2013, Wukovits
et al. 2017).
Foraminifera are unicellular organisms, which live predominantly in marine environments. A
recent field study showed that benthic foraminifera can account for up to 84% of total protozoan
biomass in mudflats (Lei et al. 2014). Many foraminifera feed on phytoplankton (algae,
diatoms) and thus play an important role in passing on energy in form of organic matter to
higher trophic levels (Azam et al. 1983, Beringer et al. 1991). Due to the large quantity of
foraminifera in the deep and shallow ocean waters and their large contribution to the uptake of
primary produced organic material, foraminifera significantly contribute to the global marine
carbon and nitrogen cycles (Altenbach 1992, Graf 1992, Gooday et al. 1992, Nomaki et al. 2008,
Glock et al. 2013).
Foraminifera can even change between active feeding and passive ingestion diets
depending on how much food is available (Sliter 1965). Some foraminifera can retain organelles
(chloroplasts) from certain food sources and integrate them into their own metabolic cycle. This
process is commonly referred to as kleptoplastidy. Currently nine benthic foraminiferal genera
are known to follow this lifestyle: *Bulimina*, *Elphidium*, *Haynesina*, *Nonion*, *Nonionella*,
*Nonionellina*, *Reophax*, *Stainforthia* und *Virgulinella* (Lopez 1979, Lee et al. 1988, Cedhagen
1991, Bernhard & Bowser 1999, Correia & Lee 2000, Grzymski et al. 2002, Goldstein et al
2004, Pillet et al. 2011, Lechliter 2014, Tsuchiya et al. 2015). In the temperate Wadden Sea,
being a part of the North Sea, two foraminifera species occur most frequently, *Ammonia tepida*
and *Haynesina germanica*, and have been relatively well studied in terms of trophic ecology.
While *Ammonia* does not seem to be able for kleptoplastidy (Jauffrais et al. 2016), *H. germanica*
possesses chloroplasts which are absorbed from food (microalgae) and are retained as organelles
(Lopez 1979). Cesborn et al. (2017) demonstrated that the plastids in *H. germanica* are
photosynthetically active, based on changes in $O_2$ consumption rates during dark-light
transitions. *Haynesina germanica* therefore follows a mixotrophic lifestyle, with autotrophic
and heterotrophic nutrition (Cesborn et al. 2017). While *Ammonia* can rapidly ingest organic
carbon (Moodley et al. 2000) and *A. tepida* has a higher potential to convert algal organic matter
into cellular biomass in a short time frame compared to *H. germanica* (Wukovits et al. 2018),
the latter species (*H. germanica*) can eventually reduce its dependency on external food due to
the presence of kleptoplasts.
The uptake of food by foraminifera depends on several factors such as food quality and
quantity, temperature and salinity (Lee et al. 1966, Dissard et al. 2009, Wukovits et al. 2017).
Past experiments with *A. tepida* and *H. germanica* showed that increasing temperature had a
negative effect on food uptake of foraminifera (Wukovits et al. 2017). Highest food uptake rates
were recorded at 20 °C. As the temperature increased foraminifera of both species consumed
less food (Wukovits et al. 2017). Today not only increasing temperature but also salinity
changes play an important role in the oceans, mainly because of anthropogenic influence,
however effects of salinity on food uptake and digestion by foraminifera have not yet been
studied. Based on the strong variability and fluctuations in salinity levels in the intertidal
systems we studied food uptake of *A. tepida* and *H. germanica* at different salinity levels to





provide a better understanding of the turnover of phytoplankton by foraminifera with changing
physical conditions (salinity).

**2. Materials and Methods**

2.1. Sampling
The sample material was collected in May 2018 during low tide at Friedrichskoog Spitze
(German Wadden Sea, at 54° 02' N, 8° 50' E). At that time the seawater had a salinity of 24.2
PSU and a temperature of 13 °C, and the air temperature was 11 °C. The collected sediment
was directly wet-sieved at the site through a 125 and a 63 µm sieve to remove larger meiofauna
and smaller organic particles. In the laboratory, the sediments (size class 63-125 µm) containing
living benthic foraminifera were fed regularly with *Dunaliella tertiolecta* (green algae) until the
start of the experiment and were kept at a temperature of 21 °C and a salt content of 24 PSU. 1
PSU (1 practical salinity unit) corresponds approximately to 1 g salt per kg seawater.

2.2. Preparation of $^{13}C^{15}N$-labeled phytodetritus
A f/2 medium (Guillard & Ryther 1962, Guillard 1975), enriched with $^{13}C$ (1.5 mmol
$NaH^{13}CO_3/L$) and $^{15}N$ (0.44 mmol $Na^{15}NO_3/L$) was used as a nutrient solution for the cultivation
and production of isotopically labeled *D. tertiolecta*, a common food source in laboratory
experiments with benthic foraminifera (e.g. Heinz et al. 2002, Wukovits et al. 2017). It should
be noted that *H. germanica* prefers to eat diatoms (Austin et al. 2005), however significant
uptake of *D. tertiolecta* was also previously reported (Wukovits et al. 2017). The algal culture
was kept in an incubator at 20 °C with a light/dark cycle of 16:8 h. Once the algae had grown
to high density in the medium, they were collected by centrifugation at 800 x*g* for 10 minutes.
The algal pellet was washed three times with artificial seawater (Enge et al. 2011). After each
washing step the culture was centrifuged and the supernatant decanted. For the storage of the
labelled algae, the pellet was shock frozen in liquid nitrogen and then lyophilized for 4 days at
0.180 mbar. The labeled algal powder was isotopically enriched at about 3.3 at%$^{13}C$ and 32.3
at%$^{15}N$.

**3. Sample preparation and analysis**

3.1. Sample preparation
The experiment was run in triplicates. For each salinity level (11, 24 and 37 PSU) and each time
point of harvest (1, 5 and 14 days) three glass crystallization dishes were setup for *A. tepida* and
for *H. germanica*. The selected salinities correspond to a brackish milieu (11 PSU), to the
natural conditions in the North Sea (24 PSU) and to a highly saline basin (37 PSU). For *A.*
*tepida* 55 individuals and for *H. germanica* 60 individuals were prepared per replicate to obtain
a dry mass of cytoplasm between 1 and 2 mg. The crystallization dishes were filled with 280 ml
of filtered natural seawater from the sampling site. The salinity was previously adjusted to the
desired PSU value by adding NaCl or distilled water. The foraminifera were then placed in the
dishes (without sediment) and acclimated at 20 °C and a light/dark cycle of 16:8 h for three days





in an incubator. After the acclimation period, 5 mg lyophilized labelled algal powder was added
as the only food source to each replicate and left in the incubator for the desired incubation time.
In addition, untreated foraminifera were taken to obtain the natural abundance of $^{13}$C and $^{15}$N as
a reference. At the end of the experiments a precipitate of the algal powder was still visible in
the crystallization dishes, which confirms the continuous availability of food during the
experiments. The salinity was checked daily and corrected when necessary.

3.2. Sample preparation and processing
Before the start of the experiments all glassware was cleaned by combusting at 500 °C for 5 h
in a muffle furnace. The „picking tools" and tin capsules were cleaned by rinsing with a 1:1
(v:v) mixture of dichloromethane ($CH_2Cl_2$) and methanol ($CH_3OH$). After the incubation period,
foraminifera were removed from the crystallization dishes, cleaned and washed three times with
distilled water. Then they were transferred into the tin capsules (Sn 99,9%, IVA
Analysentechnik GmbH & Co. KG) and excess water was removed. The samples were air dried
for three days (Enge et al. 2018) and then decarbonated with 4% HCl (3 x 5 µL for *A. tepida*
and 2 x 5 µL for *H. germanica*). During the decarbonatization of foraminiferal tests, the samples
were kept at 60 °C for 24 h. Finally, the samples were dried for three days at 60 °C, before being
weighed to the nearest hundredth of a milligram.

3.3. Analyses
The measurements of C and N contents as well as the isotope ratios of the samples were carried
out in the Stable Isotope Laboratory for Environmental Research (SILVER) laboratory of the
University of Vienna. The ratios of $^{13}$C/$^{12}$C and $^{15}$N/$^{14}$N were measured by an isotope ratio mass
spectrometry (IRMS, Delta$^{PLUS}$, coupled by a ConFlo III interface to an elemental analyzer EA
1110, Thermo Finnigan). In the following calculations, X stands for the heavy isotopes of C and
N, i.e. $^{13}$C and $^{15}$N, respectively. The atomic percentage of heavy isotopes (at%$^{13}$C and at%$^{15}$N)
was calculated using the measured $\delta^{13}$C and $\delta^{15}$N values and the international standards for C
(Vienna PeeDee Belemnite $R_{VPDB}$ = 0.0112372) and N isotopes (atmospheric nitrogen $R_{atmN}$ =
0.0036765) according to the following equations:

$\delta X = (R_{sample}/R_{standard} - 1) \times 1000$ (1)

where R depicts the ratio of heavy isotope to light isotope i.e. $^{13}$C:$^{12}$C or $^{15}$N:$^{14}$N in samples and
international standards, respectively.

$$\text{at. \%} = \frac{100 \times R_{standard} \times (\frac{\delta X_{sample}}{1000} + 1)}{1 + R_{standard} \times (\frac{\delta X_{sample}}{1000} + 1)}.$$

159 (2)


Subsequently, the values needed to be corrected for the at%X present in the natural
environment, i.e. in unlabeled foraminifera. The so-called isotope excess (*E*) was calculated
according to Middelburg et al. (2000):




$$E = \frac{\text{atom}X_{\text{sample}} - \text{atom}X_{\text{background}}}{100}$$

(3)


In the next step, the isotope incorporation was determined according to the following equation:

$I_{\text{iso}}$ [µg mg$^{-1}$] or [µg ind$^{-1}$] = E x C (N) [µg mg$^{-1}$] or [µg ind$^{-1}$]  (4)

Depending on the biomass units used, $I_{iso}$ results in the unit µg mg$^{-1}$ (based on dry matter of the
cytoplasm) or µg ind$^{-1}$ (based on the number of individuals).

Finally, the uptake of phytodetrital C (pC) and phytodetrital N (pN) was calculated for the
cytoplasm of foraminifera:

$$pX = \frac{I_{\text{iso}}}{\frac{\text{at.} \% X_{\text{phyto}}}{100}}$$

(5)


where at%$_{\text{phyto}}$ represents the isotopic enrichment in $^{13}$C and $^{15}$N of the labelled *D. tertiolecta*
food. All results were additionally converted to time-based food uptake rates (µg mg$^{-1}$ h$^{-1}$).

3.4. Statistics
Regression analysis was applied to statistically test for time effects on food uptake, and linear
and curvilinear models were tested. The best models were selected based on the highest
coefficient of determination ($R^2$). Three-way analysis of variance (ANOVA) was applied to test
for main effects of species, salinity and time, and two-way ANOVA for salinity and time effects
on pC and pN within species, followed by Fisher's LSD post hoc tests. All statistical tests were
performed using R (R development Core Team, 2008).

**4. Results**

4.1. Carbon uptake
The isotope measurements showed that the offered labeled food source was utilized by both, *A.*
*tepida* and *H. germanica*. Three-way ANOVA showed a significant effect of species (*A.*
*tepida* > *H. germanica*, *p<0,001*), time (*p<0,001*) and salinity *(p<0,001)* on pC. Moreover, two-
way ANOVA highlighted a significant effect of time *(p<0,001)* and salinity *(p<0,001)* on pC
in *A. tepida*, and of time *(p<0,001)* but not salinity *(p=0,0739)* on pC in *H. germanica*. Salinity
had a major impact on food uptake (pC) only in *A. tepida*.
As shown in Fig. 1A, *A. tepida* had the highest pC value at a salinity level of 37 PSU
for the most dates, followed by 24 PSU. At lowest salinity (11 PSU) pC further decreased. It
should be noted that from day 1 to day 5 the uptake of C at 24 PSU and 37 PSU decreased
considerably before it increased again towards day 14. This intermediate minimum was not





recognizable at 11 PSU. At 11 PSU pC increased linearly with time (f(d) = 0.05163*d +
0.06530, $R^2$=0.9985, based of mean values of pC).

Time kinetics were different for *H. germanica*. After one feeding day the measured pC

values did not differ between salinity levels and were lowest. Food C uptake peaked after five
days and thereafter declined. However, salinity did not affect pC in this species.

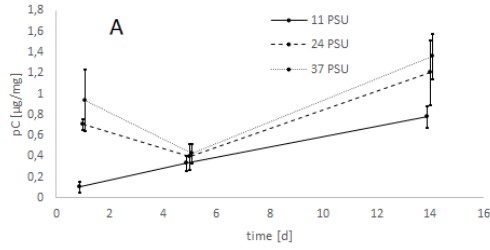


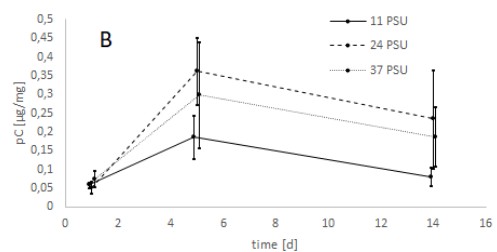

Figure 1: Time kinetics of algal C uptake (pC) by (A) *A. tepida* and (B) *H. germanica*. pC was measured at three salinity
levels: 11, 24 and 37 PSU.

In addition to pC values, C uptake rates were also determined (Figure 2). *Ammonia tepida*
showed highest uptake rates at 24 and 37 PSU after one day of food supply and exponentially
decreasing rates afterwards. For 11 PSU, C uptake rates were more or less stable over time
(Figure 2). For *H. germanica,* C uptake rates at salinities of 11 and 37 PSU followed an almost
linear trend (decrease). At 24 PSU, C uptake rates increased from day 1 to 5 and then declined
towards day 14.




Figure 2: Carbon uptake rates of *A. tepida* (A) and *H. germanica* (B). *Ammonia tepida* followed a typical exponential
decrease, whereas *H. germanica* showed a more linear decrease.

4.2. Nitrogen uptake
Two-way ANOVA showed a significant effect of salinity ($p<0,001$) and time ($p<0,001$) on
nitrogen uptake (pN) for *A. tepida*. For *H. germanica,* as with pC, pN was only affected by time
($p=0,0027$) but not by salinity ($p=0,0690$).

Nitrogen uptake of *A. tepida* showed a highly comparable pattern to C uptake (Figure

3A). Minimum N uptake was always recorded at the lowest salinity level. However, the uptake
of N after 5 days was approximately the same at 24 and 37 PSU, and reached here a minimum
at both salinities. The development of pN at 11 PSU could be described by a straight line (f(d)
$= 0.02354*d + 0.02011$) with a very high coefficient of determination ($R^2 = 0.9978$).

*Haynesina germanica* exhibited lower values of pN compared to *A. tepida* (Figure 3B).

The highest N uptake after 5 and 14 days was at the moderate salinity level (24 PSU), though
this was not significant. Again food N uptake increased linearly with time (f(d) = $0.00185*d +$
$0.03522$, $R^2 = 0.9317$) at the lowest salinity level, but showed a saturating behavior at 24 and
37 PSU.



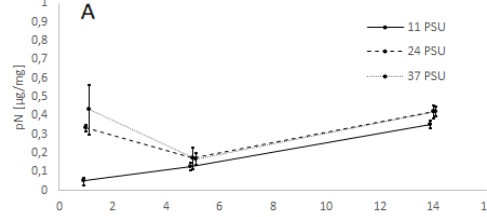


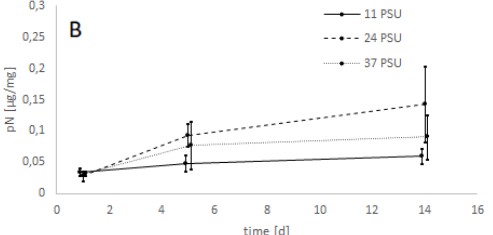






Figure 3: Time kinetics of algal N uptake (pN) by (A) *A. tepida* and (B) *H. germanica*. pN was measured at three salinity
levels: 11, 24 and 37 PSU.

Food N uptake rates are shown in Figure 4. For *A. tepida* the N uptake rates developed similar
to the C uptake rates i.e. they declined exponentially over time (24 and 37 PSU) and C uptake
rates were approximately twice as high as N uptake rates (Figure 4A). In *H. germanica* large
differences between the C and the N uptake rates were observed (Figure 4B). The time kinetics
of N uptake rates were no longer linear but decreased exponentially at all salinity levels.
Furthermore, the average N uptake rates were very close at all three salinity levels, suggesting
similar N uptake rates independent of salinity in *H. germanica*.

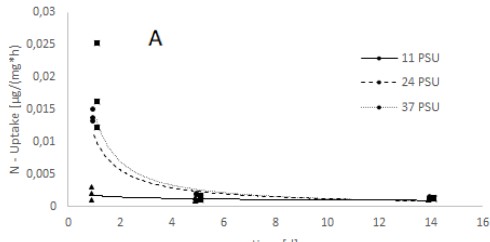


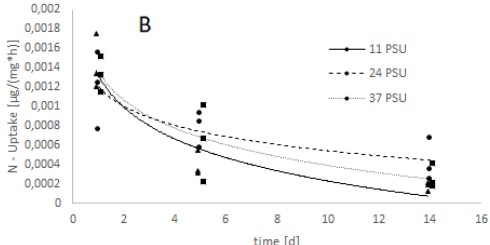


Figure 4: N uptake rates of *A. tepida* (A) and *H. germanica* (B). Both species showed an exponential decrease in N
uptake rates over time. The triangles correspond to the values at 11 PSU, the circles to those at 24 PSU, and the squares
to the values at 37 PSU.

4.3. Relations between food C and N incorporation

All data of C and N uptake obtained in this study were plotted as pC to pN relationships in
Figure 5.

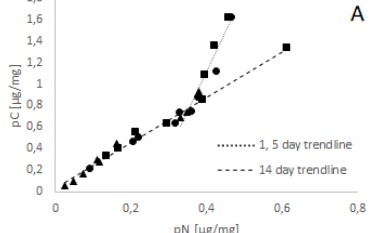







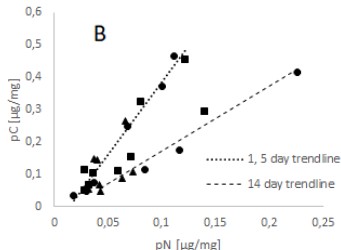



Fig. 5: Relationship between food N uptake (pN) and food C uptake (pC) for *A. tepida* (A) and *H. germanica* (B) for
the time windows day 1 to 5, and day 14. Regressions were run separately for these time windows. The triangles
correspond to the values at 11 PSU, the circles to those at 24 PSU, and the squares to the values at 37 PSU.

*Ammonia tepida* showed a continuous increase in pC with pN (Fig. 5). The 1-day and 5-day
samples were all plotting on a straight line, with the lowest salinity samples having the lowest
pC and pN values. At later stages (day 14) the slope and therefore the pC:pN ratios increased
markedly (from 2.1 to 7.4). *Haynesina germanica* also showed a general increase in pC with
pN. However, the slope between both of them decreased over time in contrast to *A. tepida*,
indicating a decrease in pC:pN (from 4.5 at day 1 and 5 to 2.0 at day 14) and thereby an increased
relative retention of food N compared to food C over time.

**5. Discussion**

5.1. Influence of salinity on food uptake
Both examined foraminifera species showed different responses to salinity variations in terms
of food uptake and food uptake rates. The time course of pC and pN in *A tepida* showed a
noticeably minimum after five days. This partial decrease in pC and pN was already reported in
experiments testing the effects of temperature on food uptake in the same species (Wukovits et
al. 2017). In the latter study food uptake was highest on day one and then decreased sharply (5
days) and remained nearly constant thereafter (14 days). These data suggest that *A. tepida* was
„starved" due to the 3-day acclimatization period and immediately responded with rapid food
uptake, when food was added. The pseudopodia of *A. tepida* are particularly stimulated by the
green algae *Dunaliella* (Lee et al 1961). Excessive food uptake in the short time (1 day) can
lead to longer lasting saturation, which explains the significantly lower uptake rates at the
intermediate time points.
The time course of food uptake at the lowest salinity level was different in *A. tepida*,
starting slow but then pC and pN increased continuously over time. This might be caused by
lowest salinity levels being suboptimal in the short term and that therefore metabolic
activation takes longer, causing the linear increase in pC and pN. This explanation supported
by the observation that after five days food uptake was similar across all three salinity levels.
*Haynesina germanica* showed a different pattern than *A. tepida* in terms of time-dependency
of pC and pN. In the former species the presence of kleptoplasts may have attenuated the
„starvation effect", with the result that only a small amount of ingested C and N can be



measured after one day. This low initial C and N uptake can be related to the results of
Cesborn (2017), which show that kleptoplasts are potential C or N sources for foraminifera in
starvation periods. However, it should be considered that *H. germanica* less readily absorbed
the offered food compared to *A. tepida*. Although there was a greater increase in pN between
day 1 and 5 than between day 5 and 14, the C and N uptake rates were much lower than those
of *A. tepida*.

It must be noted that while food C and N uptake are related through the C:N of the food

source, internal foraminiferal metabolism and release processes can cause a decoupling of C
and N metabolism and of isotope patterns. Carbon is incorporated into organic molecules as
well as into the calcareous shells or simply released during cellular respiration (e.g. Hannah et
al 1994). The latter leads to a release of carbon into the environment, whereby the measured
values of C isotope incorporation are influenced. Nitrogen is also utilized for the production of
organic molecules such as DNA or proteins (DeLaca 1982, Nomaki et al 2014). Again the
release of nitrogen-rich excretion products into the environment has an impact on the nitrogen
isotope incorporation patterns.

Experiments by Stouff et al. (1999) showed that *A. tepida* has hardly any anomalous

shell formation at normal marine conditions of 37 PSU. This observation is consistent with the
results of this study, as *A. tepida* had a higher uptake and turnover of organic matter at higher
salinities (24 – 37 PSU) and therefore its optimal living conditions at higher salinity levels.
Yet, in the hypersaline environment (50 PSU) this species generates a high number of
deformed juvenile individuals (Stouff et al. 1999). The German Wadden Sea is subject to
seasonal salinity fluctuations and has a mean salinity of 30.7 – 32.5 PSU (Postma 1983).
Depending on the supply of fresh water and evaporation rates, the water in this region can
drop to salinities of 25 and reach up to 37 PSU (Maywald 1991). Our experiments showed that
the change in salinity from 24 to 33 PSU had a smaller impact on food uptake than that
between 11 and 24 PSU. This shows once again that the two commonly occurring species, *A.*
*tepida* and *H. germanica*, have adapted very well to these fluctuations. The lowest salinity (11
PSU) in our experiments represents the transition from brackish to a marine milieu. It turned
out that at this salinity level the food uptake tended to be the lowest for both species. From the
literature it is known that such brackish marshes are mainly inhabited by agglutinated
foraminifera (Sen Gupta 1999). Considering the uptake of C and N by *A. tepida* and *H.*
*germanica* in our experiments (Fig. 1, 3), it can be seen that the low salinities do not
correspond to the optimum conditions of these foraminifera.

5.2. Effect of salinity on cytoplasmic C:N ratios and $\delta^{13}C$ values
Foraminiferal C:N ratios and $\delta^{13}C$ signatures in the cytoplasm have been applied as a salinity
proxy for marine systems for some time (e.g. Scott and Medioli 1986. Chmura and Aharon
1995. Mackie et al. 2005). According to Mackie et al. (2005) $\delta^{13}C$ values in the range of -16 to
-22‰ represent organic matter and organisms of marine origin. Brackish and freshwater
organisms have higher $\delta^{13}C$ values (-22 to -25‰ and -25 to -30‰ respectively) (Mackie et al.
2005). The foraminiferal species studied here showed background $\delta^{13}C$ values of -13.9‰ (*H.*
*germanica*) and -15.9‰ (*A. tepida*). These values clearly point towards marine isotope



344 signatures, concordant with a salinity of 24.2 PSU measured during the sampling of the

345 foraminifera.

346 A change in cytoplasmic C:N ratio of foraminifera in intertidal habitats is fundamentally

347 influenced by two factors: on the one hand by the composition of the local fauna and flora (food)

348 and on the other hand by changes in the physiological processes in the organisms themselves

349 (Frost und Elser 2002, Stelzer und Lamberti 2001, Bowman et al 2005, Cross et al 2005,

350 LeKieffre 2018). Both benthic foraminifera species showed divergent changes in C versus N

351 metabolism of ingested food over time. *Ammonia tepida* showed an increase in pC:pN with

352 feeding time, resulting from a combination of altered N metabolism (storage of N in form of

353 proteins or DNA versus N excretions) and/or changes in C metabolism (investment of C into

354 cellular components versus losses by cellular respiration). The observed increase in pC:pN may

355 therefore represent either an increase in C incorporation relative to N incorporation due to lower

356 stress (less cellular respiration) or a decrease in N retention (increased N excretion) in the

357 foraminifera after a prolonged feeding time. *Haynesina germanica* also showed a general

358 increase in pC with pN. However, the slope between pC and pN decreased over time, indicating

359 a decrease in pC:pN and thereby an increased relative retention of food N compared to food C.

360 In our experiments the change in salinity did not affect the pC:pN ratios. In other words the

361 salinity did not cause a change in relative C versus N metabolism in both species. Investigating

362 the behavior of other nutrients such as P or Mg alongside C and N might provide further

363 interesting insights into the intake and metabolism of food and its biochemical constituents.

364 Phosphorus serves as an important building block in nucleic acids and phospholipids and might

365 be an indicator for cellular energy status because it is used for the formation of energy storage

366 molecules such as ATP. The behavior of P at changing environmental conditions may therefore

367 indirectly indicate the stress behavior of foraminifera. Magnesium is an important component

368 of chlorophyll. Based on the Mg content of foraminifera it is possible to reconstruct the amount

369 of chlorophyll and therefore the presence of chloroplasts. However, this is only possible if the

370 pure cytoplasm is examined without the residues of the shells.

371   An important point is the different affinity of foraminifera to food. As *H. germanica*

372 possesses kleptoplasts, which are absent in *A. tepida*, the two species have different metabolisms

373 and food dependencies. *Ammonia tepida* showed an approximately 10-fold higher food uptake

374 as *H. germanica*, partially explained by the preference of *A. tepida* for the green algae

375 *Dunaliella sp*. (Lee et al 1961) which served as the food source here while *H. germanica* prefers

376 to eat diatoms due to kleptoplastidy.

377   Furthermore. the alteration and aging of food sources can play an important role

378 affecting feeding and food metabolism, as indicated by the preference for „fresh" or

379 „younger" phytodetritus (Lee et al 1966). In the experiments here food from the same

380 lyophilized algal batch was always used to avoid this effect. Moreover, selective food uptake of

381 different species of foraminifera needs to be considered, and this was clearly demonstrated in a

382 study where a total of 28 different diatom and chlorophyte species were fed to three littoral

383 benthic foraminifera species but only 4-5 of these food sources were consumed at significant

384 rates (Lee and Müller 1973). Ultimately one needs to be aware that contamination by bacteria

385 or other microbes cannot be ruled out, particularly in longer-term experiments, as these



organisms also use the food offered as a C or N source (Murray et al 1986. Dobbs et al 1989.
Middelburg et al. 2000. Gihring et al 2009).
5.3. Effects of salinity on the foraminiferal community
The foraminifera of the mudflats of Friedrichskoog have been investigated for their
responses to environmental parameters such as temperature and organic matter flux (Llobret-
Brossa et al. 1998. Brasse et al. 1999. Tillmann et al. 2000). In this study we could show that *A.*
*tepida* and *H. germanica* reacts with a lower food uptake compared to a decreasing salinity. At
low tide the benthic organisms are strongly exposed to the ambient weather conditions such as
wind, rain or sun. Due to the geographic location the growth of organisms is strongly linked to
the spring and summer months. Past data from Tillmann et al. (2000) showed that growth of
phytoplankton in winter is limited or almost zero. During spring local phytoplankton blooms
may occur with a daily water column particulate gross production up to 2200 mg C m$^{-2}$ day$^{-1}$
(Tillmann et al. 2000). Over this period food availability is not a limiting factor for foraminifera
and this situation corresponds to the conditions in our experiments.
The composition of the foraminiferal community in the German Wadden Sea changes
within small areas (subzones) (Müller-Navarra et al. 2016). The specific microhabitats are
formed by natural parameters such as sediment grain size, pH or food source availability but
also by anthropogenic influences such as diking, ditching or sheep grazing (Müller-Navarra et
al. 2016). This leads to changes in the hydrological situation, and in combination with natural
factors such as precipitation or seepage of ground water, the salinity in mudflats varies
significantly in relation to the open ocean (De Rijk 1995). It seems that the assemblage of
foraminifera in such human-influenced salt marshes is controlled mainly by changes in salinity
(De Rijk 1995). De Rijk (1995) showed that in areas with widely varying salinity only few
different types of foraminifera occur. Moreover, it was shown that in years with high
precipitation the salinity in areas such as the Wadden Sea or in salt marshes is reduced, causing
the density of foraminifera to decrease sharply. (Murray 1968). So the tidal habitats in the region
around Friedrichskoog are characterized by multiple environmental factors. This leads to the
formation of subzones, where particularly physical influences such as pH, salinity, temperature
or tides play an important role. This area is also of particular interest for the future as the
anthropogenic impact on fluctuating ecosystems can be monitored very well here. Changes in
salinity therefore are a major factor shaping the composition and activity of foraminiferal
communities. In this study we could show that the two tested foraminiferal species, *A. tepida*
and *H. germanica*, responded very differently to salinity in terms of food intake and C and N
metabolism. Moreover, a former study demonstrated that the temperature response and
temperature optima also differ between these two most abundant foraminifera species of the
German Wadden Sea (Wukovits et al. 2017). Therefore environmental and climate change can
strongly affect the composition of the foraminiferal community, thereby causing changes in the
feeding rates and in the C-N metabolism of the foraminiferal community, and ultimately altering
the C-N cycling of these intertidal ecosystems.




## 6. Literature

Allen J.: Morphodynamics of Holocene salt marshes: a review sketch from the Atlantic and southern North Sea coast of Europe; Quaternary Science Reviews. v. 19. pp: 1155-1231, 2000.

Altenbach A.: Short-term processes and patterns in the foraminiferal response to organic flux rates. Mar Micropaleontol 19:119–129, 1992.

Austin H., Austin W., Paterson D.: Extracellular cracking and content removal of benthic diatom *Pleurosigma angulatum* (Quekett) by the benthic foraminifera *Haynesina germanica* (Ehrenberg). Mar. Micropaleontol.. 57. 68-73, 2005.

Azam F., Fenchel T., Field J., Gray J., Meyer-Reil L. and Thingstad F.: The ecological role of water-column microbes in the sea. Mar. Ecol. Prog. Ser., 10:257-263, 1983.

Beringer U., Caron D., Sanders R., and Finaly B.: Heterotrophic flagellates of planktonic communities. their characteristics and methods of study. In: Patterson D. J. & Larsen. J. (ed.). The Biology of Free-Living Heterotrophic Flagellates. Vol. 45. Clarendon Press. Oxford. p. 39-56, 1991.

Bernhard J. and Bowser S.: Benthic foraminifera of dysoxic sediments: chloroplast sequestration and functional morphology: Earth-Science Reviews. v. 46. p. 149–165, 1999.

Brassea S., Reimerb A., Seiferta R., Michaelis W., The influence of intertidal mudflats on the dissolved inorganic carbon and total alkalinity distribution in the German Bight. southeastern North Sea; ElSEVIER. Volume 42. Issue 2. pp:93-103, 1999.

Bowmann M., Chambers P., Schindler D.: Changes in stoichiometric constraints on epilithon and benthic macroinvertebrates in response to slight nutrient enrichment of mountain rivers; Freshwater Biology 50. doi 10.1111/j.1365-2427.2005.01457.x., 2005.

Caldeira K.. Wickett M.: Ocean model predictions of chemistry changes from carbon dioxide emissions to the atmosphere and ocean. J. Geophys. Re., 110. C09S04. doi:10.1029/2004JC002671, 2005.

Cedhagen T.: Retention of chloroplasts and bathymetric distribution in the Sublittoral Foraminiferan *Nonionellina labradorica*: Ophelia. v. 33. p. 17–30, 1991.

Cesborn F., Geslin E., Kieffre E., Jauffrais T., Nardelli M., Langlet D., Mabilleau G., Jorissen F., Jezequel D. and Metzger E.: Sequestered Chloroplasts in the benthic Foraminifer *Haynesina germanica*: Cellular organization. oxygen fluxes and potential ecological implications. Journal of Foraminiferal Research v. 47. no. 3. p. 268-278, 2017.

Chmura G., Ahron P.: Stable carbon isotope signatures of sedimentary carbon in coastal wetlands as indicators of salinity regime. Journal of Coastal Research. 11: 124-135, 1995.

Cross W., Benstead J., Frost P. and Thomas S.: Ecological stoichiometry in freshwater benthic systems: recent progress and perspectives; Freshwater Biology 50. pp: 1895-1912, 2005.

Correia M. and Lee J.: Chloroplast retention by *Elphidium excavatum* (Terquem). Is it a selective process?: Symbiosis. v. 29. p.343–355, 2000.





DeLaca T.: Use of dissolved amino acids by the foraminifer *Notodendrodes antarcticos*; Amer.
Zool. 22:683-690, 1982.

De Rijk S.: Salinity control on the distribution of salt marsh foraminifers (Great Marshes.
Massachusetts). Journal of Foraminiferal Research v. 25. pp: 156-166), 1995.

Dissard D., Nehrke G., Reichart G. and Bijma J.: The impact of salinity on the Mg/Ca and
Sr/Ca ratio in the benthic foraminifera *Ammonia tepida*: Results from culture
experiments. Geochimica et Cosmochimica Acta 74 (2010) 928-940, 2009.

Dobbs F., Guckert B. and Carmin K.: Comparison of three techniques for administering
radiolabeled substrates to sediments for trophic studies: Incorporation by microbes;
Microbiol. Ecol. 17. 237-250, 1989.

Enge A., Nomaki N., Ogawa N., Witte U., Moeseneder M., Lavik G., Ohkouchi N., Kitazato
H., Kucera M. and Heinz P.: Response of the benthic foraminiferal community to a
simulated short-term phytodetritus pulse in the abyssal North Pacific. Mar. Ecol.-
Prog. Ser. 438. 129-142, 2011.

Enge A., Wanek W. and Heinz P.; Preservation effects on isotopic signatures in benthic
foraminiferal biomass; Marine Micropaleontology; Volume 144, Pages 50-59, 2018.

Frost P. and Elser J.: Effect of light and nutrients on the net accumulation and element
composition of epilithon in boreal lakes; Freshwater Biology 47. pp: 173-183, 2002.

Glock N., Schönfeld J., Eisenhauer A., Hensen C., Mallon J. and Sommer S.: The role of
benthic foraminiferain the benthic nitrogen cycle of the Peruvian oxygen minimum
zone; Biogeoscience 10, 4767-4783, 2013.

Goldstein S., Bernhard J. and Richardson E.: Chloroplast Sequestration in the Foraminifer
*Haynesina germanica*: Application of High Pressure Freezing and Freeze
Substitution: Microscopy and Microanalysis. v. 10. p. 1458–1459, 2004.

Gooday A.: The role of benthic foraminifera in deep-sea food webs and carbon cycling. In:
Rowe GT. Pariente V (eds) Deep-sea food chains and the global carbon cycle.
Kluwer Academic Publishers. Dordrecht. p 63–91, 1992.

Graf G.: Benthic-pelagic coupling: a benthic review. Oceanogr Mar Biol Annu Rev 30:149–
190, 1992.

Grzymski J., Schönfield O., Falkowski P. and Bernhard J.: The function of plastids in the
deep-sea benthic foraminifer. *Nonionella stella*. Limnology and Oceanography. v. 47.
p. 1569–1580, 2002.

Gihring T., Humphrys M., Mills H., Huet M. and J. Kostka: Identification of phytodetritus-
degrading microbial communities in sublittoral Gulf of Mexico sands; Limnology and
Oceanography 54. 1073-1083, 2009.

Guillard R.: Culture of phytoplankton for feeding marina Invertebrates. In: Culture of marine
invertebrates animals. Springer 29-60, 1975.

Guillard R. and Ryther J.: Studies of marina planktonic diatoms: I *Cyclotella nana* Hustedt.
and *Detonula confervacea* (CLEVE) Gran. Can. J. Microbiol. 8. 229-239, 1962.

Hannah F., Rogerson R. and Laybourn-Parry J.: Respiration rates and biovolumes of common
benthic Foraminifera (Protozoa); Cambridge University Press. Volume 72. Issue 2. pp
301-312, 1994.



Heinz P., Hemleben C. and Kitazato H.: Time-response of cultured deep-sea benthic
foraminifera to different algal diets; Oceanographic Research Papers Vol. 49, Issue 3,
pp: 517-537, 2002.
Jauffrais T., Jesus B., Metzger E., Mouget J., Jorissen F. and Geslin E.; Effect of light on
photosynthetic efficiency of sequestered chloroplasts in intertidal benthic
foraminifera (*Haynesina germanica* and *Ammonia tepida*); Biogeosciences. 13. 2715-
2726, 2016.
Keul N., Langer G., Nooijer L. and Bijma J.: Effect of ocean acidification on the benthic
foraminifera *Ammonia* sp. is caused by a decrease in carbonate ion concentration.
2013.
Lechliter S.: Preliminary study of kleptoplastidy in foraminifera of South Carolina: Bridges. v.
8. p. 44, 2014.
Lee J., Price S., Tentchoff M. and McLaughin J.: Growth and Physiology of Foraminifera in
the Laboratory: Part 1: Collection and Maintenance – Micropaleontology Vol. 7. No.
4 pp. 461-466, 1961.
Lee J., McEnery M., Pierce S., Freudenthal H. and Müller W.: Tracer experiments in feeding
littoral foraminifera; J. Protozool. 13, 657-670, 1966.
Lee J., Lanners E. and Kuile B.: The retention of chloroplasts by the foraminifer *Elphidium
crispum*: Symbiosis. v. 5. p. 45–59, 1988.
Lee J. and Müller W.: Trophic dynamics and niches of salt marsh foraminifera; Am. Zool. 13.
215-223, 1973.
Lei Y., Stumm K., Wickham S. and Beringer U.: Distributions and Biomass of Benthic
Ciliates. Foraminifera and Amoeboid Protists in Marine. Brackish. and Freshwater
Sediments. J. Eukaryot. Microbiol. 61. 493-508, 2014.
LeKieffre C., Jauffrais T., Geslin E., Jesus B., Bernhard J. M., Giovani M. and Meibom
A.; Inorganic carbon and nitrogen assimilation in cellular compartments of a
benthic kleptoplastic foraminifer; Scientific Reports volume 8.
Article number: 10140, 2018.
Lopez E.: Algal chloroplasts in the protoplasm of three species of benthic foraminifera:
taxonomic affinity. viability and persistence. Marine Biology. v. 53. p. 201–211,
1979.
Lobet-Brossa E., Rossello – Mora R. and Aman A.: Microbial Community Composition of
Wadden Sea Sediments as Revealed by Fluorescence in Situ Hybridization;
Environmental Microbiology Vol. 64. No. 7, 1998.
Maywald A.: Das Watt. 1. Aufl. Ravensburg: Maier, 1991.
Mackie E., Leng M., Lloyd J. and Arrowsmith C.: Bulk organic d13C. C/N ratios as
paleosalinity indicators within a Scottish isolation basin; Journal of Quaternary
Science 20: 303–312, 2005.
Middelburg J., Barranguet C., Boschker H., Herman P., Moens T. and Heip C.: The fate of
intertidal microphytobenthos carbon: An in situ $^{13}$C-labeling study. Limnol. Oceanogr.
45. 1224-1234, 2000.





Moodley L., Boschker H., Middelburg J., Pel R., Herman P. and De Deckere E.: Ecological
significance of benthic foraminifera [13]C labelling experiments. Mar Ecol Prog Ser
Vol. 202:289-295, 2000.
Murray J.:The living Foraminifera of Christchurch Harbour, England: Micropal. V. 14, no.4,
p:83-96, 1968.
Murray R., Cooksley K. and Priscu J.: Stimulation of bacterial DNA synthesis by algal
exudates in attached algal-bacterial consortia; Appl. Environ. Microbiol. 52. 1177-
1182, 1986.

Müller-Navara K., Milker Y. and Schmiedl G.: Natural and anthropogenic influence on the
distribution of salt marsh foraminifera in the bay of Tümlau. German North Sea;
Journal of Foraminiferal Research 46. pp 61-74, 2016.
Nomaki H., Ogawa N., Ohkouchi N., Suga H., Toyofuku T., Shimanaga M., Nakatsuka T. and
Kitazato H.: Benthic foraminifera as trophic links between phytodetritus and benthic
metazoans: carbon and nitrogen isotopic evidence; MEPS 357:153-164, 2008.
Nomaki H., Chikaraishi Y., Tsuchiya M., Ohkouchi N., Uematsu K., Tame A. and Kitazato H.:
Nitrate uptake by foraminifera and use in conjunction with endobionts under anoxic
conditions; Limnology and Oceanography Volume 59. Issue 6. pages 1879-1888,
2014.

Pillet L., Vargas C. and Pawlowski J.: Molecular identification of sequestered diatom
chloroplasts and kleptoplastidy in foraminifera: Protist. v. 162. p. 394–404. 2011
Postma H.: Hydrography of the Wadden Sea: Movements and properties of water and
particulate matter. In: WOLFF, W. J. (Hrsg.): Ecology of the Wadden Sea. A. A.
Balkema, Rotterdam: 2/1-2/75, 1983.
Sen Gupta B.: Foraminifera in marginal marine environments. Modern Foraminifera 141 – 159,
1999.

Schafer C., Cole F., Frobel D., Rice N. and Buzas M.: An in situ experiment on temperature
sensitivity of nearshore temperate benthic foraminifera. Oceanographic Literature
Review. 9. 913, 1996.
Scott F. and Medioli D.: Foraminifera as sea-level indicators. Sea Level Research. pp: 435-
456, 1986.

Stelzer R. and Lamberti G.: Effect of N:P ratio and total nutrient concentration on stream
periphyton community structure. biomass and element composition; Limnology and
Oceanography 46. pp:356-367, 2001.
Stouff V., Geslin E., Debenaj J. and Lesourd M.: Origin of morphological abnormalities in
Ammonia (Foraminifera): studies in laboratory and natural environments. Journal of
Foraminiferal Research 29 (2). 152-170, 1999.
Sliter W.: Laboratory Experiments on the Life Cycle and Ecological Controls of *Rosalina*
*globularis* d'Orbigny. Journal of Eukaryotic Microbiology Volume 12. Issue 2.
pages 210-215, 1965.
Tillmann U., Hesse K. and Colijn F.: Planktonic primary production in the German Wadden
Sea; Journal of Plankton Research. Volume 22. Issue 7. Pages 1253–1276, 2000.



Tsuchiya M., Toyofuko T., Uematsu K., Bruchert V., Collen J., Yamamoto H. and Kitazato
H.: Cytologic and genetic characteristics of endobiotic bacteria and kleptoplasts of
*Virgulinella fragilis* (Foraminifera): Journal of Eukaryotic Microbiology. v. 62. p.
454–469, 2015.
Wukovits J., Enge A., Wanek W., Watzka M. and Heinz P.: Increased temperature causes
different carbon and nitrogen processing patterns in two common intertidal
foraminifera (*Ammonia tepida* and *Haynesina germanica*). Biogeosciences 14. 2815-
2829, 2017.
Wukovits J., Oberrauch M., Enge A. and Heinz P.; The distinct roles of two intertidal
foraminiferal species in phytodetrital carbon and nitrogen fluxes – results from
laboratory feeding experiments; Biogeoscience. 15. 6185-6198, 2018.