# Peer review of "Salinity-depending algae uptake and subsequent carbon and nitrogen"

_Biogeosciences, 2019_

## Referee Comment (RC1) · Anonymous Referee #1 · 1 Oct 2019

Lintner et al. reported about the experimental results to investigate salinity effects on foraminiferal feeding activities using two foraminiferal species from German Wadden Sea. They found different responses to salinity between species, partly attributing to their salinity adaptation and kleptoplasty. The effects of salinity on foraminiferal feeding have not been reported abundantly, so thier results add interesting insights. The manuscript is generally clear and well written, however, I suggest two major points to revise before its acceptance.

1. Food uptake rates. Line 214: Because foraminifera excrete ingested C and N with time, it is not appropriate to divide pC with time in particular for long period incubation,

as authors discussed in lines 308 to 316. The "exponentially decreasing rate" may be due to "apparent" decrease of pC due to excretion/mineralization. I strongly suggest to compare C and N uptake rates between different salinity only between 1 day after.

2. According to Fig. 1, pC accounts only ∼0.1% of the biomass of Ammonia and much less for H. germanica. It is important to add some discussions to compare these values to other studies, and speculate why the uptake rate is so low in this experiments.

Other specific comments.

Title: There may be no discrimination of C and N during food uptake, but occur during assimilation/metabolisms afterwards. I therefore suggest to change "Salinity-depending algae uptake and subsequent carbon and nitrogen metabolisms of two...".

Lines 74 to 75: Please indicate temperature ranges they tested.

Line 94: Why do you specifically state as "salt content of 24PSU" rather than "salinity of 24"?

Line 108: Why did the authors freeze-dried the isotopically labeled algae? Was "frozen and thawed before offering to foraminifera" not appropriate? If so, why??

Line 124: Did you mix the lyophilized algae with seawater before its addition?

Line 135: I think washing with distilled water makes damages on foraminiferal cytoplasm by osmotic shock. Please mention potential effects of this to the discussion.

Line 229: How about to plot N uptake next to C uptake (Figs 3 and 1 together), making easier comparison between them.

Line 275: Please indicate the C:N ratios of Dunaliella as well. This is very important information for interpretation.

Line 310: metabolism and of isotope "labeling" patterns.

Line 317: Is this sentence grammatically correct?

Line 326: 37, not 33?

Line 338: Are these authors actually used d13C as a proxy of salinity? Or did they use d13C to estimate the origin of OM, either marine phytoplankton or terrestrial plants??

Line 341: Lighter or lower d13C values.

Line 349: It is better to separate citations into two parenthesis; one after "fauna and flora (food)" and the other after "organisms themselves".

Line 367: I would quantify Chlorophyll directly rather than Mg quantification, although Mg is of course important element to understand foraminiferal physiology, in particular responses to salinity changes.

Line 371: New chapter from this paragraph?

Line 377: Furthermore","

395: Sunlight?

---

## Referee Comment (RC2) · Anonymous Referee #2 · 19 Feb 2020

General comments:

The manuscript submitted by Lintner et al. describes the results of salinity-depending feeding experiments with the two most common foraminiferal species in the mud flats Ammonia tepida and Haynesina germanica. As far as I know, the linking between food uptake and salinity has rarely been assessed by other authors and therefore, the new findings of the authors of this article contribute to a better understanding of the physiological behaviour of foraminifers, which could potentially support proxy applications. Generally, the manuscript is well structured, clearly written and I would recommend publishing the manuscript after reassess some minor points/ comments.

Main point:

Line 93: How did you know that the foraminifera where alive before and after the experiment (e.g. Pseudopodial activity, crawling test, maybe some kind of staining living cells?)? It would be nice if you could go more into detail! How many foraminifera did survive the experiment? Following my culturing experience, it never happens that all specimens survive. Do you see any differences in survival rates between the different salinities?

General Questions concerning culturing method:

1: Have you done any water exchange during the 14 day culturing period?

2: Have you measured or monitored the oxygen content of the water?

3: How was counteracted against evaporation? By adding distilled water? How much evaporation occurred over the day? Have you closed the culturing vessels with a lid or something?

4: Have you measured the pH?

Specific comments:

Line 91: What kind of water have you used for wet-sieving? Artificial seawater, water from the location?

Line 94 and 104: Why have you decided to culture under such a high temperature? Because of the highest food uptake rates mentioned in Wukovits et al. 2017? The annual mean temperature of the water is definitely much lower and especially in spring when the highest abundance of food is given temperatures are lower.

Line 121: What was the pore size of the filter?

Line 125: Have you suspended the food in artificial water or anything else before adding?

[Figure]

Line 126: What do you mean with "untreated"? Are the foraminifera not fed at all in the same setting of the experiment or are they directly picked from the stock culture?

Line 141: Is it possible that residue of clay minerals from the original sediment where still in or around the foraminifera?

Line 377: Furthermore,

Line 400: The situation does only correspond to the conditions in your experiment in terms of food but not in terms of temperature. The temperature in spring will be much lower.

―――――――――――――――――――――――――

---

## Author Comment (AC1) · 9 Mar 2020

ad Anonymous Referee #1

major points:

1: We agree with your suggestion– only values of the 1d samples are now compared.

2: There are several explanations for that listen here and we included them in the discussion. - An important aspect to consider is the method used when processing the samples. Foraminiferal tests are dissolved with hydrochloric acid and due to that carbonate is lost, but also new mineral phases are formed which influence the total

weight of the sample. This step is needed to remove the 13C, which may be bound in the test. Comparing with other studies like Wukovits et al. 2017, it can be seen, that the uptake values of our experiments lay in the same order of magitude. - Another aspect is, that foraminifera are stressed during experimental conditions and therefore may have a lower turnover. It should be noted, that the food uptake is determined by the isotope content in the cytoplasm, which, as recommended in major point 1, can also vary over time. - Since these experiment were all carried out under laboratory conditions, we would not consider pC and pN as absolute values, due to seasonal and environmental fluctuations. However, a difference in food uptake can be shown in our experiments with varying salinity.

Specific comments:

Title: Good point, title has been changed.

Lines 74-75: We added the tested temperature range (line 77).

Line 94: The meaning is the same; it only serves to avoid repetition. But we changed it now and wrote "salinity of 24" to avoid confusion.

Line 108: In the method used here, the algae were processed in the "most gentle" way, which means that no further degradation of algae or gross damage of the cells occured. During the "normal" freezing, the algae could be decomposed by further microbacterial activities or burst and therefore lose cytoplasm, which would lead to a change of the 13C and 15 N content.

Line 124: No, the algae powder was added directly into the crystallization dishes and mixed there. The algae settled down on the bottom of the dish and were available for foraminifera. We added this information now.

Line 135: Potential effect was added to the discussion. In general, all samples were always treated the same way, which means they were all washed with the same volume of distilled water. This way any impact that may have arisen from using the distilled

water has the same effect on all samples.

Line 229: It has been changed and all graphs are plotting in Fig. 1 now.

Line 275: The content of at%13C and at%15N was added in line 109. This helps to calculate the atomic C:N ratio. We added this information.

Line 310: "Labeling" was added.

Line 317: We corrected the sentence.

Line 326: 33 is the salinity used in the experiments. 37 PSU are extreme values, that Maywald (1991) has found in the North Sea.

Line 338: They used d13C to estimate the origin of OM and as a consequence they could differentiate between a brackish and marine milieu. Below, an example is described in detail, that can be directly related to our experiments (Mackie et al. 2005).

Line 341 and 349: We improved the text here in accordance to the reviewer.

Line 367: We added this information.

Line 371: We would suggest to keep the structure here and add only interesting aspects, that have not been mentioned yet. In our opinion, a new chapter would disrupt the flow of reading.

Line 377: . was replaced with ,

Line 395: We improved the text here.

---

## Author Comment (AC2) · 9 Mar 2020

ad Anonymous Referee #2

Main point:

Line 93: For our experiments, we only used foraminifera with densely filled cytoplasm. In addition, we only used individuals with an intense yellowish color of the cytoplasm. The incubation time of foraminifera in the crystallization dishes before feeding was used for the "crawling test". Foraminifera were placed in the center of the crystallization dish immediately after removal from the cultures. After 24 hours individuals could be

identified that have moved away from the center. Accordingly, these individuals have active pseudopodia and are alive. After finishing the experiments, the individuals were also checked for the intense yellowish color, but no "crawling test" has been carried out, since additional stress and time (with continuing breathing and excretion) could strongly affect the isotope content and therefore the results. In all our experiments is was very rare (below 4%), that single individuals did not show colored cytoplasm and they were therefore counted as "survive". Completely empty tests, which would clearly stand for dead individuals, were not found. We added this information to the methods.

General Questions:

1, 2 & 4: In order not to disturb the experiments, no water change was carried out. O2 and pH were also not measured, since we did not expect a significant change due to the small amount of added food.

3: The crystallization dishes were sealed tightly with parafilm. The water level and regular control of the salinity showed, that there was no evaporation. We added this information to the methods.

Specific comments:

Line 91: We used water from the location. We added this information to the methods.

Line 94 and 104: ad 94: The 21 °C correspond to the room temperature of the laboratory, where the foraminiferal cultures were placed. Ad 104: the 20°C refer to the temperature in the incubator, where algae were cultivated. Experience shows, that algae grow very well at 20°C. The feeding experiments were also carried out at 20°C in the incubator, for optimal food uptake conditions. This experience is based on the experiments of Wukovits et al. (2017).

Line 121: 0.45 $\mu$m – We added this information to the manuscript.

Line 125: No, the food was directly put into the crystallization dishes and mixed there. See response to Referee #1.
Line 126: "Untreated" means not fed. Foraminifera were removed directly from the stock culture and processed. We improved this sentence to avoid confusion here.

Line 141: It is difficult to say what is inside of the foraminifera. But before foraminifera were further processed, they were cleaned with a brush to remove any organic or inorganic residues that were visible on their test.

Line 377: . was replaced with ,

Line 400: We agree with the reviewer. However, in this chapter we only refer to the availability of food.